# The Epithelial-Mesenchymal Transcription Factor Slug Predicts Survival Benefit of Up-Front Surgery in Head and Neck Cancer

**DOI:** 10.3390/cancers13040772

**Published:** 2021-02-12

**Authors:** Herbert Riechelmann, Teresa Bernadette Steinbichler, Susanne Sprung, Matthias Santer, Annette Runge, Ute Ganswindt, Gabriele Gamerith, Jozsef Dudas

**Affiliations:** 1Department for Otorhinolaryngology, Head and Neck Surgery, Medical University of Innsbruck, 6020 Innsbruck, Austria; herbert.riechelmann@i-med.ac.at (H.R.); matthias.santer@tirol-kliniken.at (M.S.); annette.runge@tirol-kliniken.at (A.R.); jozsef.dudas@i-med.ac.at (J.D.); 2Institute of Pathology, Neuropathology and Molecular Pathology, Medical University of Innsbruck, 6020 Innsbruck, Austria; susanne.sprung@i-med.ac.at; 3Department of Therapeutic Radiology and Oncology, Medical University of Innsbruck, 6020 Innsbruck, Austria; ute.ganswindt@i-med.ac.at; 4Department of Hematology and Oncology, Medical University Innsbruck, 6020 Innsbruck, Austria; gabriele.gamerith@tirol-kliniken.at

**Keywords:** head and neck cancer, epithelial–mesenchymal transition, snail family transcription factors, biomarker, prognosis, proportional hazards models

## Abstract

**Simple Summary:**

In preclinical studies, the epithelial-to-mesenchymal transition (EMT)-related transcription factor Slug indicated radio- and chemoresistance in head and neck squamous cell carcinoma (HNSCC). Here we show that Slug is a biomarker associated with treatment failure in HNSCC patients treated with primary radio- or radiochemotherapy, but not in patients undergoing upfront surgery and postoperative radio- or chemoradiotherapy. Slug may thus serve as a predictive biomarker to identify HNSCC patients who will benefit from upfront surgery. Slug status is an immunohistochemical (IHC) parameter that is easy to determine. If the predictive value observed here can be confirmed in validation studies with independent data, Slug immunohistochemistry may have significant clinical relevance in treatment planning for HNSCC patients.

**Abstract:**

EMT promotes radio- and chemotherapy resistance in HNSCC in vitro. As EMT has been correlated to the transcription factor Slug in tumor specimens from HNSCC patients, we assessed whether Slug overexpression predicts radio- and chemotherapy resistance and favors upfront surgery in HNSCC patients. Slug expression was determined by IHC scoring in tumor specimens from patients with incident HNSCC. Patients were treated with either definitive radiotherapy or chemoradiotherapy (primary RT/CRT) or upfront surgery with or without postoperative RT or CRT (upfront surgery/PORT). Treatment failure rates and overall survival (OS) were compared between RT/CRT and upfront surgery/PORT in Slug-positive and Slug-negative patients. Slug IHC was positive in 91/354 HNSCC patients. Primary RT/CRT showed inferior response rates (univariate odds ratio (OR) for treatment failure, 3.6; 95% CI, 1.7 to 7.9; *p* = 0.001) and inferior 5-year OS (univariate, *p* < 0.001) in Slug-positive patients. The independent predictive value of Slug expression status was confirmed in a multivariable Cox model (*p* = 0.017). Slug-positive patients had a 3.3 times better chance of survival when treated with upfront surgery/PORT versus primary RT/CRT. For HNSCC patients, Slug IHC represents a novel and feasible predictive biomarker to support upfront surgery.

## 1. Introduction

Head and neck squamous cell carcinoma (HNSCC) is a debilitating and often fatal disease. Recent advances in the treatment of HNSCC, including minimally invasive surgical techniques, advances in radiotherapy, and multimodal approaches, have led to substantial improvements in outcomes [1]. For incident early-stage HNSCC, surgery or radiation therapy (RT) can be applied as a single treatment modality. In locally advanced stages, upfront surgical therapy with postoperative radiotherapy with or without concurrent systemic therapy (PORT) or, alternatively, primary definitive chemoradiotherapy (CRT) is a standard first-line treatment modality [1].

Several host factors, including patient fitness and preferences, and various disease factors, including disease site, stage, and resectability, are considered in a decision regarding the choice of clinical treatment [2,3]. In this context, the question of whether upfront surgery should be part of first-line therapy frequently arises. Surgical treatment poses specific additional risks for the patient. However, in about one-fourth of patients with HNSCC, primary RT/CRT fails and the disease persists, often requiring distressing salvage treatments [4,5]. Moreover, persistent disease is associated with an increased risk for distant metastasis [6] and poor survival [4]. Therefore, a predictive biomarker that identifies patients who have a high risk for failure of primary RT/CRT and who might benefit from upfront surgery is highly desirable [3,7]. So far, various tumor-intrinsic factors for predicting response to RT/CRT treatment have been studied. These include proteins related to the DNA damage response, the relative numbers of cancer stem cells, tumor hypoxia, and the p16 status, which indicates the human papillomavirus (HPV)-related etiology [8,9,10]. However, HPV/p16 status is more prognostic than predictive because survival advantages are similar between primary RT/CRT and upfront surgery/PORT [11,12,13]. TP53 mutations have been associated with radioresistance in some but not all studies [14,15]. Expression of the epidermal growth factor receptor was found to be inconclusive with regard to predicting radioresistance [16]. Currently, there is no validated predictive biomarker that not only predicts radio- or chemosensitivity but also indicates the potential superiority of upfront surgery in HNSCC patients [3,17,18].

With this background, markers indicating epithelial-to-mesenchymal transition (EMT), a phenotypic switch of epithelial cells to mesenchymal cells, are of particular interest. EMT is characterized by the cellular loss of epithelial proteins, including cytokeratins, E-cadherin, and ß-catenin, and the gain of mesenchymal proteins, such as vimentin, fibronectin, and N-cadherin [19]. EMT is a transcriptionally regulated process that promotes tumor progression, invasion, and metastasis. The associated transcription factors include zinc finger proteins SNAI1 (Snail) and SNAI2 (Slug), zinc finger E-box-binding homeobox 1 and 2 (ZEB1 and ZEB2), and Twist-related proteins 1 and 2 (TWIST1 and TWIST2) [20]. EMT and stemness are closely related [19,21]. In vitro, the induction of EMT in HNSCC cell lines doubled cisplatin resistance and increased radioresistance with a dose-modifying factor of 2 compared with the unmodified cell line [22,23]. EMT-related gene signatures are associated with radio- and chemotherapy resistance in patients with HNSCC [24]. We recently reported that EMT can be quantified in tumor samples from HNSCC patients using multichannel fluorescence image cytometry by evaluating the co-expression of epithelial and mesenchymal proteins in tumor cells [25]. The EMT-related transcription factor Slug was found to correlate with the relative count of tumor cells in EMT. Based on these image cytometry data, HNSCC samples could be divided into Slug-positive and Slug-negative tumors using a cutoff of 10% Slug-positive tumor cells [25]. No Slug at all was detectable by immunohistochemistry (IHC) in 43/47 pharyngeal control tissues obtained during surgery for snoring or sleep apnea, and low expression was observed in the remaining control tissues (Figure 1).

As Slug overexpression indicates the presence of EMT and EMT is associated with radio- and chemotherapy resistance in vitro, we determined whether Slug expression in the primary tumor can predict response to primary RT/CRT in HNSCC patients. Moreover, we assessed its predictive value, whether upfront surgery/PORT is favorable in this setting. A factor is considered predictive when it interacts with the treatment in terms of the outcome [26]. Therefore, the core question was whether the Slug IHC status interacted significantly with the outcome of first-line therapy (primary RT/CRT vs. upfront surgery/PORT) in HNSCC.

## 2. Materials and Methods

The procedures used were in accord with the Declaration of Helsinki of 1975 and its revision in 1983. Approval was obtained from the Ethics Committee of the Medical University of Innsbruck (Reference Number UN3678). Written informed consent was obtained from all included patients. The present study followed the Transparent Reporting of a multivariable prediction model for Individual Prognosis or Diagnosis (TRIPOD) statements [27]. Moreover, it took into account suggestions for the clinical validation of prognostic and predictive biomarkers [7,26,28].

### 2.1. Study Population

Patients with newly diagnosed primary squamous cell carcinomas of the head and neck treated in curative intent with RT, surgery, or CRT between January 2008 and January 2020 at the Department of Otorhinolaryngology, Head and Neck Surgery, Medical University of Innsbruck, were evaluated for available Formalin-Fixed Parrafin-Embedded (FFPE) tissues from the primary tumor site to stain for their expression of Slug. Patients’ outcomes were prospectively documented within our database, including treatment modality and response. The exclusion criteria were non-squamous head and neck cancers, other entities, cancer of unknown primary (CUP), metastatic disease, palliative treatment intent, and no tumor-specific treatment based on patient decision or frailty (Figure 2).

### 2.2. Host Factors, Disease Factors, and First-Line Treatment Modality

The recorded host factors included gender, age in years, and the American Society of Anesthesiology (ASA) physical status score as a simple measure of comorbidity [29]. ASA scores were dichotomized into ASA I/II and ASA III/IV. Further host factors included smoking history (≤10 vs. >10 pack-years) [2]. Disease-related factors included tumor site, which was grouped into oral cavity, oropharynx, hypopharynx, larynx, and other sites. The Union for International Cancer Control (UICC)-TNM (UICC-TNM) staging that was valid at the time of the initial diagnosis was used. Clinical T stage was dichotomized into T1–T3 and T4, and N was classified as either N1, N2, or N3 with no further subclassification.

Treatment modality was recommended by an interdisciplinary head and neck tumor board in line with National Comprehensive Cancer Network (NCCN) Guidelines and included upfront surgical resection, radiotherapy, and combinations with systemic treatments (Appendix A). Surgical procedures included transoral laser microsurgery (TLM), transfacial or transcervical tumor resections, pedicled or free-flap reconstructions, and uni- or bilateral selective or comprehensive neck dissections. Upfront surgery was always carried out with the aim of achieving a tumor-free margin of at least 5 mm around the formalin-fixed, paraffin-embedded tumor sample [30]. For TLM of pharyngeal and laryngeal tumors, the aim of the surgery was a deep clear margin of at least 2 mm and a peripheral clear margin of 3 mm [31].

RT was applied as a first-line treatment or as PORT in advanced disease if no high-risk factors were present. The latter case included involved margins or extracapsular lymph node extension [30]. RT was usually carried out in conventional fractions with 1.8–2.0 Gy daily, five times a week. The total dose in the area of an untreated primary tumor or in the region of untreated primary lymph node metastases was 70–72 Gy. In regions associated with a high risk of existing or postoperatively remaining tumor cells, the dose was 60 to a maximum of 66 Gy, and in areas of physiological anatomical lymphatic drainage, the dose was about 50 Gy. For postoperative patients with high-risk features and for patients with advanced disease who received primary RT, concomitant systemic therapy (CRT) was applied [30,32,33].

Concomitant systemic therapy consisted of either cisplatin, at 100 mg/m^2^ on days 1, 22, and 43 or 25 mg/m^2^ on days 1–4 and 29–32 [34], or mitomycin C, at 10 mg/m^2^ (max. 15 mg total) on days 1 and 29, and 5-fluorouracil, at 600 mg/m^2^ (24 h infusion) on days 1–5 and 29–33 [35] in patients not suitable for cisplatin treatment. Alternatively, RT was combined with cetuximab with a loading dose of 400 mg/m^2^ once a week before the start of radiotherapy followed by 250 mg/m^2^ weekly for the duration of the RT in frail patients [36].

Concomitant cetuximab and RT were grouped together with concomitant systemic therapy and RT as CRT. First-line treatment modalities were dichotomized into treatments with upfront tumor surgery with or without postoperative RT with or without concurrent systemic therapy (upfront surgery/PORT) or, in contrast, primary radiotherapy with or without concurrent systemic therapy without upfront surgery (primary RT/CRT (Appendix A). In addition to Slug IHC status (positive vs. negative), the first-line treatment modality was the main factor for analysis.

### 2.3. Treatment Response Evaluation and Follow-Up

All patients were routinely assessed for treatment response 10–12 weeks after the end of the first-line treatment [37,38,39]. The treatment response evaluation included contrast-enhanced computer tomography (CT), magnetic resonance imaging (MRI), or position emission tomography-computed tomography (PET-CT) scans and a restaging endoscopy, usually under general anesthesia with biopsies from the initial tumor sites. Treatment response was categorized according to the WHO response criteria as complete response (CR) or persistent disease, including partial response, no change, and progressive disease. For patients with first-line treatment in curative intent, persistent disease was considered a treatment failure. Relapse following a previous CR was not considered in this analysis [3]. The failure rate served as an outcome parameter in the treatment response analysis. The results of the response evaluation were presented to the interdisciplinary tumor board, which recommended second-line treatments if indicated. All patients were followed up regularly in accordance with current standards [37]. If the date of diagnosis was less than 6 years prior and the last follow-up more than 1 year prior, the patient, their relatives, or their treating physicians were contacted to obtain the survival and remission status.

### 2.4. Immunohistochemistry

All tumor samples were paraffin-embedded and sectioned, as reported earlier [40]. Immunohistochemistry for Slug was performed using the mouse monoclonal anti-Slug antibody (cat. nr. 564614, BD Pharmingen Austria, Vienna, Austria). The reaction was developed using a universal secondary antibody (Roche Ventana, Tucson, AZ, USA) and the DAB Map kit from Ventana (Figure 1). The Slug status was categorized as positive if 10% or more of tumor cells were Slug positive [25]. A commercial in vitro diagnostic certified assay containing a ready-to-use prediluted mouse monoclonal antibody was used for p16 detection (CINtec^®^ Histology V-Kit, Roche Ventana, Tucson, AZ, USA). Tumor samples were considered p16 positive if more than 60% of tumor cells showed immunohistochemical reaction products [41]. For PD-L1 IHC, the rabbit monoclonal antibody from Cell Signaling Technologies, Danvers, MA, USA, Cat. No. 13684, was used and the reaction developed as described in Slug. A positive PD-L1 reaction in 1% of cells in the tumor tissue was considered positive [42]. The immunohistochemistry of p53 was done using a commercial in vitro diagnostic certified assay containing a ready-to-use prediluted mouse monoclonal antibody (cat. nr. 760-2542, Roche Ventana). According to a previous publication by Bouchalova et al., scattered p53 staining is considered to be associated with a regular genetic background without nonsilent mutations, while no staining at all is related to complete loss of p53 protein due to deletions. An increased (over 66% of tumor cells stained) staining pattern is considered to be associated with nonsilent mutated p53. Consequently, we described p53 positivity (positive for nonsilent mutated p53) as either overexpression or total absence of p53 [43]. The IHC reaction for excision repair cross-complementation group 1 (ERCC1) was completed using a mouse monoclonal antibody (Antibodies Online, Aachen, Germany, cat. nr. ABIN 197720), for carbonic anhydrase 9 (CAIX) using a rabbit polyclonal antibody (Novus Biologicals, Centennial, CO, USA, cat. nr. NB100-417SS), and these reactions were also developed by using a universal secondary antibody (Roche Ventana) and the DAB Map kit from Ventana. Tumors were considered ERCC1 and CAIX positive if more than 30% of the cells showed positive staining [25].

In all staining procedures, isotype-matching control immunoglobulins were used in the same final concentration as in the antibody staining conditions. The isotype controls never presented any visible reactions.

### 2.5. Data Analysis

For metric data, means and standard deviations are provided unless stated otherwise. Metric variables were compared using *t*-tests or ANOVA unless stated otherwise. Frequency data, including failure rates, were tabulated and compared using chi-square or Fisher’s exact tests. Median follow-up time was calculated using the reverse Kaplan–Meier method [44]. For univariate analyses of overall survival (OS), life tables (actuarial survival) with time intervals of 12 months were used to calculate survival rates, and the Kaplan–Meier product limit method was used for the univariate analysis of cumulative survival probability. Factors were compared with the logrank tests. For the multivariable survival analysis, a Cox regression model with indicator factor coding was used. The first category served as a reference. Likelihood ratio tests were used to calculate *p*-values. Patient age and all variables listed in Table 1 except smoking history were included in the Cox model. No further variable selection was used. The proportional hazard assumption was checked with scaled Schoenfeld residuals vs. time using the function cox.zph in the R package “Survival” with the Kaplan–Meier transformation option and Efron correction for ties [45]. The score test of each covariate by time was used to test the proportional hazard assumption. To analyze the predictive value of Slug IHC status, the interaction term of Slug IHC status (positive/negative) and first-line treatment modality (upfront surgery/PORT vs. primary RT/CRT) was tested. This was the main endpoint of the analysis [26]. For internal model validation, Harrell’s c-index was calculated. Moreover, 1000 bootstrap samples were drawn for internal validation and to estimate the extent of overfitting. Statistical analyses were performed using SPSS 26 (IBM Corporation, Armonk, NY, USA); the R package survival, v3.2-3 [45]; and MedCalc statistical software, version 19.2.1 (MedCalc Software Ltd., Ostend, Belgium).

## 3. Results

### 3.1. Study Population

Between 2008 and 2019, 1124 patients with incident squamous cell carcinoma of the head and neck were treated at the Department of Otorhinolaryngology, Head and Neck Surgery, Medical University of Innsbruck. The Slug IHC of the primary tumor could be performed in tumor samples from 354 patients (Figure 2). The mean age (±SD) of these patients was 63 ± 10 years. Most patients were male (286/354; 81%; Table 1). All included patients were Caucasians. The median follow-up time was 40 (95% CI, 34–46) months.

### 3.2. Slug IHC

Slug IHC was positive in 91/354 patients (25.7%). Slug-positive and Slug-negative patients were comparable in terms of age at first diagnosis (*t*-test, *p* = 0.37), gender (chi-square, *p* = 0.88), smoking history (*p* = 0.59), T stage (*p* = 0.39), N stage (*p* = 0.31), and M stage (*p* = 0.37). There was no association between Slug IHC status and first-line treatment modality (*p* = 0.68). However, Slug-positive HNSCC patients were more frequently ASA III/IV (*p* = 0.045), and the tumor was less frequently located in the oropharynx (*p* = 0.015; Table 1). Moreover, Slug-positive patients were more frequently p16 negative (rho = −0.13; *p* < 0.001; *n* = 354) but ERCC1 (rho = 0.16; *n* = 45; *p* < 0.005), CAIX (rho = 0.29; *p* < 0.001; *n* = 175), CD44 (rho = 0.18; *p* = 0.02; *n* = 160), MMP9 (rho = 0.19; *p* < 0.001; *n* = 352), and p53 (rho = 0.27; *n* = 102; *p* < 0.001) positive. Slug expression was not related to PD-L1 expression (*p* = 0.89) [25].

### 3.3. Upfront Surgery/PORT vs. Primary RT/CRT

Of the 354 included patients, 163 (46%) were treated with upfront surgery/PORT and 164 (46%) patients were treated with primary RT/CRT. Other treatments or no treatment was performed in 27 patients (8%; Figure 2). Of the patients with upfront surgery/PORT, 79 (48%) had received only surgery, 53 (33%) had received surgery and PORT only, and 31 (19%) had received surgery and postoperative CRT. Of the patients with primary RT/CRT, 21 (13%) had received RT only, and 143 (87%) had received CRT (Appendix A). Treatment selection was independent of age at diagnosis (*p* = 0.22), gender (chi-square, *p* = 0.20), ASA score (*p* = 0.16), smoking history (*p* = 0.44), and Slug IHC status (*p* = 0.68). Significant differences with respect to first-line treatment selection were observed for tumor site (*p* = 0.001), T stage (*p* < 0.001), N stage (*p* < 0.001), M stage (*p* = 0.02), and p16 status (*p* = 0.013; Appendix A).

### 3.4. Slug IHC and Treatment Response to Primary RT/CRT

Positive Slug IHC was associated with a poor response to primary RT/CRT (Figure 3). Of the 164 patients treated with primary RT/CRT, response evaluation was available in 153. The reasons for missing response evaluations were death before the end of treatment (*n* = 9) and missed response evaluation (*n* = 2; Figure 2). Following primary RT/CRT, treatment failure was observed in 29/118 (26%) patients with Slug-negative HNSCC and in 19/35 (54%) patients with Slug-positive HNSCC (OR, 3.6; 95% CI, 1.7 to 8.0; *p* = 0.001; Table 2).

### 3.5. Slug IHC and Survival

Overall, 327/354 patients were included in the survival analyses. Of the missing 27 patients, 3 did not receive any treatment at our center, and 24 did receive palliative treatment with systemic therapy alone or best supportive care (BSC) (Figure 2). Of the 327 patients included in the survival analysis, 85 (26%) were Slug positive, and 163 (49.8%) had received upfront surgery/PORT.

### 3.6. Univariate Survival Analysis

The actuarial 5-year survival rate (±standard error) of the 327 patients available for the survival analysis was 50 ± 4%. Among Slug-negative patients (242/327), survival following upfront surgery/PORT and primary RT/CRT did not differ significantly. The 5-year survival rate for Slug-negative patients following upfront surgery/PORT (*n* = 119) was 60 ± 6%, and following primary RT/CRT (*n* = 123), it was 46 ± 7% (logrank, *p* = 0.12; Figure 4a). In contrast, a highly significant survival difference between upfront surgery/PORT and primary RT/CRT was observed among Slug-positive patients (85/327). The 5-year survival rate in Slug-positive patients treated with upfront surgery/PORT (*n* = 44) was 68 ± 10%, whereas in Slug-positive patients treated with primary RT/CRT (*n* = 41), it was 25 ± 8% (*p* < 0.001; Figure 4b).

### 3.7. Multivariable Survival Analysis

Ten covariates were included in a Cox regression model of overall survival (Figure 5; Appendix A). Indicator coding was used so that hazard ratios could be interpreted as relative risks. The interaction term between first-line treatment modality and Slug IHC status in the multivariable analysis indicated the predictive value of Slug IHC status and was the main outcome parameter [26]. Complete data for all included covariates were available for all 327 patients, who had received upfront surgery/PORT or primary RT/CRT (Figure 2). The global test of proportional hazard assumptions was not significant, and neither were the score tests for the single covariates (Appendix A), suggesting that the proportional hazard assumption of the model held [45]. Moreover, the graphs of scaled Schoenfeld residuals vs. time for each covariate indicated that the proportional hazard assumption was not relevantly violated (Appendix A).

The −2 log likelihood of the Cox model was 1101.4 (chi square, 109.6) with 16 degrees of freedom (*p* < 0.001). The model revealed significant survival differences for the covariates ASA score (*p* < 0.001), p16 status (*p* = 0.026), T4 vs. T1–3 (*p* < 0.001), N3 vs. N0 (*p* < 0.001), and the main outcome parameter, the interaction between Slug expression and first-line treatment modality (*p* = 0.017; Figure 5; Appendix A). T stage was grouped into T4 vs. T1–3 because T1–T3 is limited to the organ of origin of the tumor, while T4 extends beyond organ borders. According to cancer registry practices, N is encoded separately. N0 and T1–T3 are categorized as localized disease, and any N including N0 and T4 is categorized as regional disease.

The interaction term describes the ratio by which the relative risk of the first-line treatment changed when shifting from Slug negative to Slug positive, when age, sex, tumor site, ASA score, N stage, T stage, M stage, and p16 status were held constant. The hazard ratio for the interaction term was 3.09 (95% CI, 1.23 to 7.78), meaning the relative risk of primary RT/CRT compared with that of upfront surgery/PORT in Slug-positive patients was 1.08 × 3.09 = 3.34 (95% CI, 1.47 to 7.38; *p* = 0.004). Harrell’s concordance index was 0.74 ± 0.03 (±standard error), suggesting a good model fit. Bootstrapping with 1000 samples did not suggest relevant overfitting. The bootstrap point estimate of the hazard ratio of the interaction between Slug IHC status and first-line treatment modality was 3.04 with a 95% confidence interval of 1.34 to 10.02 (*p* = 0.013).

## 4. Discussion

A common clinical decision regarding patients with resectable HNSCC is whether upfront surgery should be part of the first-line treatment or primary radio- or chemoradiotherapy should be performed. Approximately half of the HNSCC patients in this study were treated with primary RT/CRT, and the others with upfront surgery/PORT, which is in line with a previous report from Europe [39]. A biomarker for predicting outcome in these patients with two potential curative options would be of high clinical relevance. In preclinical models, EMT correlated with radiochemotherapy resistance of head and neck cancer cells [22,23]. EMT could also be detected in tumor samples from HNSCC patients by using multichannel fluorescence image cytometry by evaluating the co-expression of epithelial and mesenchymal proteins in tumor cells [25]. The percentage of Slug-positive cells correlated with the percentage of cytokeratin/vimentin double-positive cells (r = 0.41; R2 = 0.17; *p* = 0.005). In Slug-positive tumors, 4.0 ± 2.6% of tumor cells were cytokeratin/vimentin double positive compared with 1.9 ± 1.8% in Slug-negative tumors (*p* = 0.001). Furthermore, Slug-positive HNSCC specimens had a lower expression of epithelial E-cadherin (*p* < 0.05) and β-catenin (*p* < 0.05), a lower E-cadherin/β-catenin co-expression (*p* < 0.05), and a higher vimentin/cytokeratin ratio (*p* = 0.01), indicating simultaneous downregulation of epithelial markers and upregulation of mesenchymal markers consistent with EMT [25]. Other EMT-related transcriptional factors, like Twist and ZEB1, were also evaluated in control and head and neck cancer tissue biopsies. These EMT-related transcriptional factors had a low overall expression and showed no significant difference of gene expression in HNSCC related to normal mucosa Consequently, we think that Slug is the major EMT-related transcriptional factor in HNSCC, and we decided to use Slug as a clinical indicator for EMT in HNSCC in this study. Slug is a feasible biomarker for the detection of EMT in HNSCC tissue and might also serve as a predictive biomarker for radiochemotherapy resistance.

Slug overexpression, that is, immunohistochemical reaction products in more than 10% of tumor cells, was observed in 26% of HNSCC patients. It was equally frequent between patients treated with upfront surgery/PORT and those treated with primary RT/CRT. Among Slug-negative patients, the failure rate of primary RT/CRT was 25%, which is in line with current reports [4]. However, among Slug-positive patients, the failure rate of primary RT/CRT was 54%. Failure of primary RT/CRT was therefore 3.6 times more likely for Slug-positive patients than for Slug-negative patients (95% CI, 1.7 to 8.0; *p* = 0.001; Table 2). In contrast to primary RT/CRT, the CR rate after upfront surgery/PORT did not differ significantly between Slug-positive and Slug-negative HNSCC patients (*p* = 0.16). The high failure rate of Slug-positive patients treated with primary RT/CRT also impaired survival. Among patients treated with primary RT/CRT, median survival was 55 months among Slug-negative patients (*n* = 123) and only 17 months among Slug-positive patients (*n* = 41; *p* < 0.001). Among patients treated with upfront surgery/PORT, 5-year survival among 119 Slug-negative HNC patients was 60%, and among 44 Slug-positive patients, it was 68% (*p* = 0.104). Accordingly, no adverse effect of Slug overexpression on survival was found in patients who underwent upfront surgery.

The predictive value of Slug was also observed in a multivariable survival analysis. Ten standard prognostic covariates were included in a Cox proportional hazards model (Figure 5; Appendix A). Smoking was closely correlated with p16 status (*p* < 0.001), not significant (*p* = 0.17), and excluded from the model. The graph of scaled Schoenfeld residuals by time (Appendix A) suggested no severe violation of the proportional hazard assumption. Harrell’s concordance index indicated a good model fit. Bootstrapping did not indicate relevant overfitting. The results of the Cox regression are plausible. The interaction term of first-line treatment modality and Slug overexpression, the main outcome parameter [26], remained significant in the multivariable model (*p* = 0.017). The relative risk of primary RT/CRT compared with that of upfront surgery/PORT increased from 1.08 (not significant) among Slug-negative patients to 3.34 among Slug-positive patients (95% CI, 1.47 to 7.38; *p* = 0.004). This result suggests that patients with Slug-positive HNSCC should be treated with upfront surgery if possible.

The high failure rate among Slug-positive patients treated with primary RT/CRT is likely due to the high number of tumor cells in EMT, which is a characteristic associated with radio- and chemotherapy resistance [24,46] and tumor stemness [21]. In previous in-vitro studies, we observed Slug to be expressed in HNC cell lines, where EMT was induced by co-cultivation with fibroblasts [47]. Furthermore, induced EMT in HNC cell lines resulted in increased resistance to radio- and chemotherapy [22,23]. We subsequently aimed to determine how to quantify EMT in tumor samples from HNSCC patients [25]. Slug was identified as a representative and feasible biomarker of EMT. At a cutoff of 10% Slug-positive tumors cells, tumors showed a markedly higher count of cytokeratin/vimentin double-positive tumor cells, which are considered a hallmark of EMT [25]. This cutoff, which indicates a high number of cells in EMT, was a priori used in the present analysis of clinical data.

Slug overexpression was associated with other biomarkers that may promote radio or chemoresistance. Slug expression and p16-positivity were inversely correlated (*p* < 0.001), in line with a previous report [48]. P16-positive HNSCC often involves wild-type p53, and Slug overexpression was correlated with either total absence of p53 or p53 overexpression (*p* < 0.001), both indicating lack of functional p53 [43], which leads to faster Slug degradation [49]. Slug correlated positively with the scores for the Ki-67 proliferation index (*p* = 0.005).** Moreover, Slug expression correlated with the expression of CAIX (*p* < 0.001), CD44 (*p* = 0.02), MMP9 (*p* < 0.001), and ERCC1 (*p* < 0.005). Slug expression did not correlate with PD-L1 expression [25]. Data about the role of both these biomarkers in radio- or chemoresistance of HNSCC are inconsistent [50,51,52]. However, none of these Slug-associated biomarkers did even come close to the predictive power of Slug with regard to the benefit of upfront surgery.

Surprisingly, little is known about predictive factors regarding upfront surgery vs. primary RT/CRT in HNSCC. In one study, TP53 mutation was found to predict the superiority of upfront surgery over primary radiotherapy among Danish HNSCC patients [15]. Cartilage invasion was shown to predict a better response to total laryngectomy compared with radiotherapy in laryngeal carcinoma in one small study [53]. Because predictors of a better outcome of upfront surgery are rare, indicators of radio- or chemoresistance are often used instead. Such indicators have been extensively studied and can be classified into factors related to hypoxia and anemia, DNA damage repair, cell proliferation, immunosuppression, stem cells, and EMT [8,9,17,18]. However, the actual superiority of upfront surgery/PORT in the presence of such factors has not been demonstrated. For example, large tumor volumes are predictors of a poor response to primary RT/CRT [54]. We previously found that upfront surgery/PORT was not superior to primary RT/CRT in large-volume tumors [55]. Similarly, oropharyngeal carcinomas have been reported to respond particularly well to radiotherapy if they are p16 positive. It has also been shown that p16-positive oropharyngeal carcinomas respond better to primary surgical therapy [11,12,13]. Accordingly, tumor volume and p16 status are prognostic rather than predictive factors.

The main limitation of this study is the retrospective design without random treatment allocation. As a consequence, only patients with resectable tumors who were fit for surgery were included in the upfront surgery/PORT arm, whereas the primary RT/CRT arm also included patients with unresectable disease or patients unfit for surgery. However, the survival rates between patients treated with upfront surgery/PORT and those treated with primary RT/CRT were almost identical in the Cox model (Figure 5; Appendix A). Apparently, the covariates in the model were able to adjust for the differences in resectability and fitness; therefore, they may also not substantially bias the interaction term treatment*Slug IHC.

A related potential source of bias is that study inclusion was based on the availability of a result of a clinical investigation, here availability of Slug IHC. This common source of selection bias is frequently not recognized [26]. Slug IHC was performed on tumor biopsies from 354/1124 HNSCC patients treated during the observation period. Patients ≤50 years and >80 years of age and with a smoking history of less than 10 pack-years, rare tumor sites, and low T stage were underrepresented among the patients with available Slug IHC (Appendix A). However, since these characteristics appear rarely in HNSCC patients, the patients with available Slug IHC are still regarded as essentially representative.

Another major limitation of this study is that the results have not yet been validated with an independent data set [7] and should therefore be considered preliminary. It should finally be noted that in contrast to univariate [56] or genomic analyses [57], Slug IHC status was not prognostic in this multivariable analysis (*p* = 0.13; Figure 5; Appendix A).

## 5. Conclusions

Slug is a feasible IHC biomarker associated with treatment failure in HNSCC patients treated with primary radio- or radiochemotherapy, but not in patients undergoing upfront surgery and postoperative radio- or chemoradiotherapy. Slug may thus serve as a predictive biomarker to identify HNSCC patients who will benefit from upfront surgery. Slug status is an immunohistochemical (IHC) parameter. If the predictive value observed here can be confirmed in validation studies with independent data, Slug immunohistochemistry may have significant clinical relevance in treatment planning for HNSCC patients. 

## Figures and Tables

**Figure 1 cancers-13-00772-f001:**
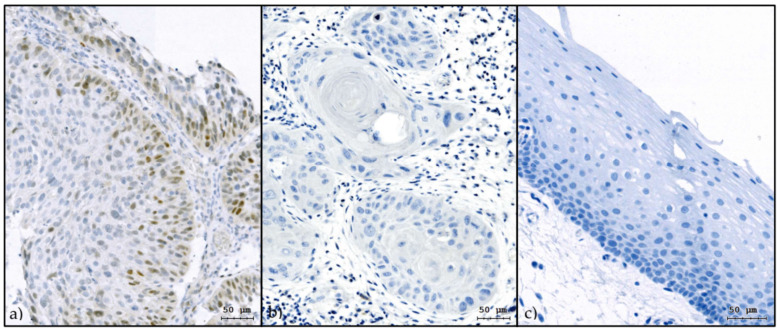
Slug expression in HNSCC and control tissue. Slug immunohistochemistry of a Slug-positive oropharyngeal carcinoma (more than 10% Slug-positive cells) (**a**), a Slug-negative oropharyngeal carcinoma (less than 10% Slug-positive cells) (**b**), and normal oropharyngeal mucosa (control) obtained during uvulopalatopharyngoplasty for sleep apnea (**c**). The HNSCC samples were p16-negative; normal oropharyngeal mucosa is, in general, Slug negative as depicted here. Only tumor cells or epithelial cells were counted; stromal or subepithelial cells were not considered. DAB with hematoxylin counterstain; bar indicates micrometer [25].

**Figure 2 cancers-13-00772-f002:**
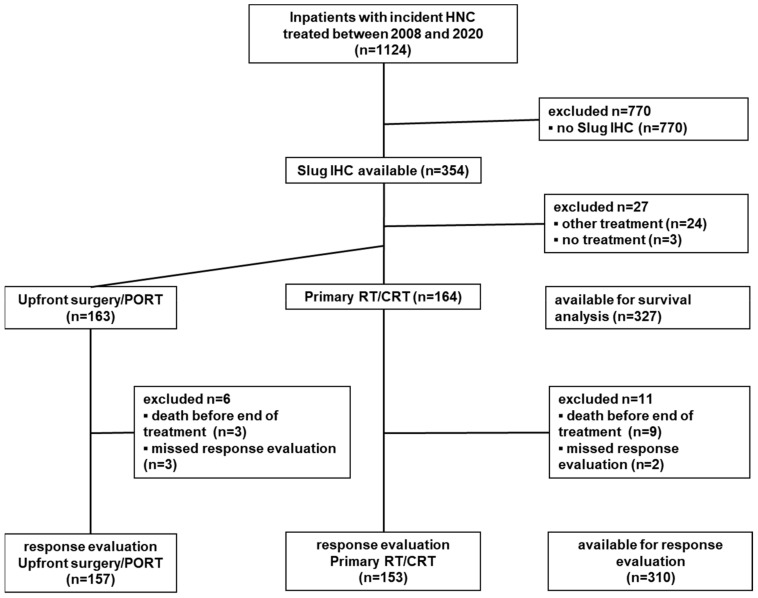
Flowchart for the selection of the study population. Between January 2008 and January 2020, 1124 inpatients with incident HNSCC were treated at the Department of Otorhinolaryngology, Head and Neck Surgery, Medical University of Innsbruck. Slug IHC status was available for tumor biopsies from 354 patients, who were included as study population. Neither upfront surgery/PORT nor primary RT/CRT was used for 27 patients, who were therefore not available for the comparison of these two treatment modalities. The remaining 327 patients were available for survival analysis. For 17 of these patients, survival status was known but not the result of treatment response analysis, leaving 310 patients for treatment response analysis.

**Figure 3 cancers-13-00772-f003:**
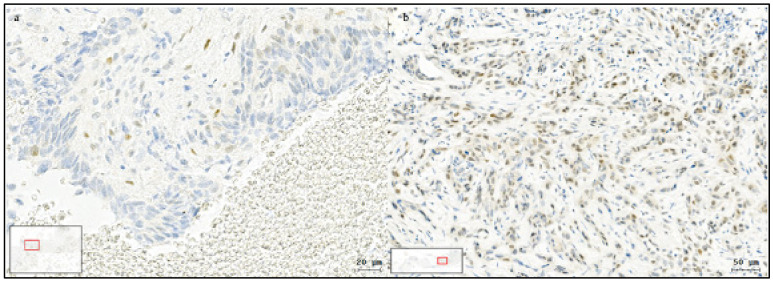
Slug expression in persistent HNSCC after first-line radiochemotherapy. Slug immunohistochemistry of a Slug-positive hypopharyngeal carcinoma after first-line radiochemotherapy. Mainly Slug-positive tumor cells survived and are surrounded by necrotic cancer tissue (**a**). Slug-positive laryngeal cancer after radiochemotherapy. Most cancer cells are Slug positive, and no necrotic cancer tissue is visible (**b**). Both patients received first-line radiochemotherapy for organ preservation but were then treated with salvage laryngectomy after initial therapy failure.

**Figure 4 cancers-13-00772-f004:**
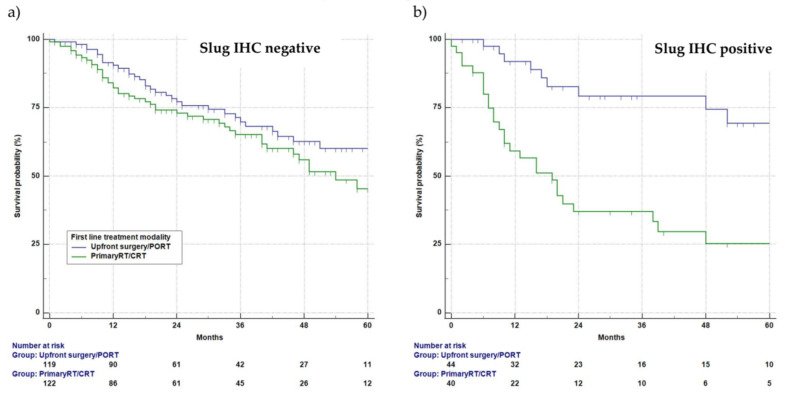
Kaplan–Meier plots of overall survival. Kaplan–Meier overall survival curves for Slug-negative (**a**) and Slug-positive (**b**) patients with incident HNSCC treated with upfront surgery/PORT (green) or primary RT/CRT (blue). For Slug-negative tumors (**a**), no significant difference in overall survival was observed according to first-line treatment modality (logrank, *p* = 0.12). For Slug-positive tumors (**b**), patients receiving upfront surgery/PORT survived significantly longer than patients treated with primary RT/CRT (*p* < 0.001).

**Figure 5 cancers-13-00772-f005:**
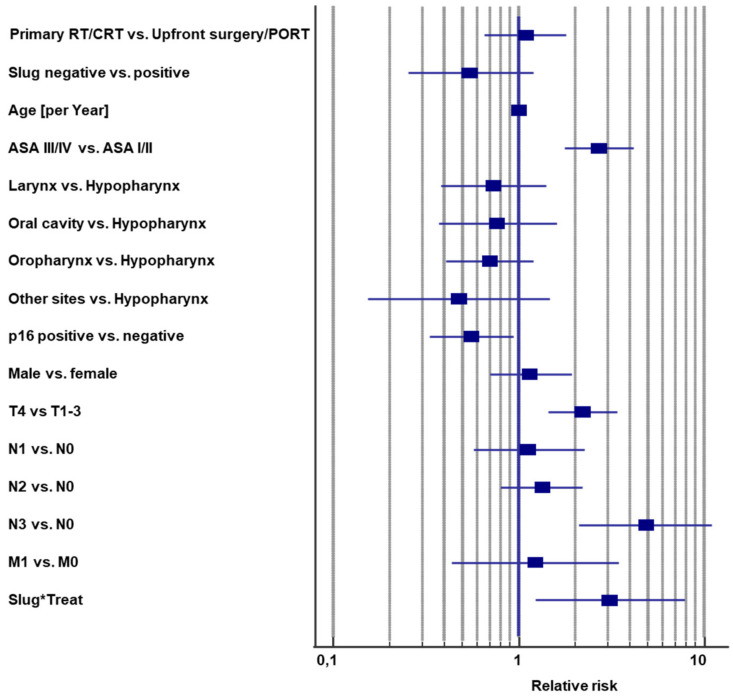
Forest plot of covariates in the Cox proportional hazards model. Point estimates and 95% confidence intervals of the relative risks for 10 covariates in a Cox regression model of overall survival. The interaction term between first-line treatment modality and Slug IHC status (Slug*Treat; relative risk, 3.09; 95% CI, 1.23 to 7.78; *p* = 0.017) indicates the predictive value of Slug overexpression for patients with incident HNSCC when corrected for the influence of the other covariates in the model.

**Table 1 cancers-13-00772-t001:** Study population. Host and disease characteristics of 354 patients with incident HNSCC and available Slug IHC status. Data are presented separately for Slug IHC status (negative, positive) and for the total study population (total). See footnotes for significant differences. Mean age at diagnosis was 63 ± 10 years and did not differ between Slug-IHC-positive and Slug-IHC-negative patients (*p* = 0.37).

Patient Factors		Slug IHC					
		Negative (*n* = 263)		Positive (*n* = 91)		Total (*n* = 354)	
		Count	Column %	Count	Column %	Count	Column %
Gender	Male	212	81%	74	81%	286	81%
	Female	51	19%	17	19%	68	19%
ASA score ^1^	ASA I/II	155	59%	43	47%	198	56%
	ASA III/IV	106	41%	48	53%	154	44%
Smoking	<10 pack-years	92	35%	29	32%	121	34%
	≥10 pack-years	171	65%	62	68%	233	66%
Tumor site ^1^	Lips and oral cavity	35	13%	16	18%	51	14%
	Oropharynx	126	48%	32	35%	158	45%
	Hypopharynx	27	10%	21	23%	48	14%
	Larynx	62	24%	17	19%	79	22%
	Others	13	5%	5	5%	18	5%
T stage (I–III vs. IV)	T1–3	186	71%	60	66%	246	69%
	T4	77	29%	31	34%	108	31%
N stage	N0	77	29%	35	38%	112	32%
	N1	55	21%	17	19%	72	20%
	N2	109	41%	35	38%	144	41%
	N3	22	8%	4	4%	26	7%
Distant metastasis	M0	248	94%	88	97%	336	95%
	M1	15	6%	3	3%	18	5%
p16 status ^2^	Negative	181	69%	81	89%	262	74%
	Positive	81	31%	10	11%	91	26%
First-line treatment modality	Upfront surgery/PORT	119	49%	44	52%	163	50%
	Primary RT/CRT	123	51%	41	48%	164	50%

^1^ Chi-square, *p* < 0.05; ^2^ chi-square, *p* < 0.001.

**Table 2 cancers-13-00772-t002:** Slug IHC status and failure rate of primary RT/CRT. Of the 354 patients included, 157 were treated with primary radiotherapy or chemoradiotherapy (primary RT/CRT) and underwent a systematic response evaluation. Of these, 35 (22%) were Slug IHC positive. The risk for failure of primary RT/CRT (i.e., persistent disease at the response evaluation 10–12 weeks after treatment) was 3.6 higher among Slug-positive patients than among Slug-negative patients (OR, 3.6; 95% CI, 1.7 to 8.0; *p* = 0.001).

Slug Status	CR	Failure	Total
Slug IHC negative	89 (75%)	29 (25%)	118
Slug IHC positive	16 (46%)	19 (54%)	35
Total	105	48	157

IHC: immunohistochemistry; CR: complete response.

## Data Availability

Data is contained within the article or Appendix A.

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
