# Peer review of "The Epithelial-Mesenchymal Transcription Factor Slug Predicts Survival Benefit of Up-Front Surgery in Head and Neck Cancer"

_cancers, 2021, doi:10.3390/cancers13040772_

Round 1
Reviewer 1 Report
The current manuscript entitled,"The epithelial-mesenchymal transcription factor Slug predicts survival benefit of up-front surgery in head and neck cancer" determined the Slug (a Zinc finger transcription factor) as an EMT predictive marker in HNSCC and treatment outcomes. Authors have performed Slug protein expression analysis by IHC in FFPE from HNSCC patients and correlated with clinical prognostic markers, RT/CRT treatment response, and OS. The study is scientifically sound and shown the expression of Slug protein to predicting the prognosis and therapeutic significance in HNSCC prevention.
I am wondering that did authors evaluated the co-expression of Slug with other epithelial/mesenchymal cell phenotype markers such as E-cadherin, Vimentin etc?
Authors need to perform IHC IgG control and normalize non-specific binding/background, which allows for better interpretation of specific staining at the antigen site.
Did the authors determine Slug expression in cancer proliferating cells with or without treatment? Authors can perform co-expression with Ki67.
Figure 1. Authors need to qualify the protein expression in IHC from Slug positive and negative samples and IgG control normalizes.
The authors need to provide a few representative IHC figures of slug expression from treated group (table 2).
Did authors check Slug expression in HNSCC provided on the TCGA portal? Which could be the best tool to compare present observation with other studies?
Author Response
Object: Manuscript ID: cancers-1095802; Type of manuscript: Article; Title: The epithelial-mesenchymal transcription factor Slug predicts survival benefit of up-front surgery in head and neck cancer; Authors: Herbert Riechelmann, Teresa Bernadette Steinbichler *, Susanne Sprung, Annette Runge, Ute Ganswindt, Gabriele Gamerith, Jozsef Dudas; Received: 16 January 2021
Author’s response to review
Thank you for consideration of our manuscript for publication in your journal.
We have reviewed the manuscript according to the reviewer’s and the editor’s comments carefully.
Reviewer #1 (Comments to the Author (Required)):
The current manuscript entitled, "The epithelial-mesenchymal transcription factor Slug predicts survival benefit of up-front surgery in head and neck cancer" determined the Slug (a Zinc finger transcription factor) as an EMT predictive marker in HNSCC and treatment outcomes. Authors have performed Slug protein expression analysis by IHC in FFPE from HNSCC patients and correlated with clinical prognostic markers, RT/CRT treatment response, and OS. The study is scientifically sound and shown the expression of Slug protein to predicting the prognosis and therapeutic significance in HNSCC prevention.
Thank you for you thorough revisions of our manuscript. We appreciate the time you spent reviewing our manuscript and the positive comments. We answered all your comments and performed the revisions accordingly.
I am wondering that did authors evaluated the co-expression of Slug with other epithelial/mesenchymal cell phenotype markers such as E-cadherin, Vimentin etc?
We have evaluated the co-expression of Slug with other markers of EMT in a previously published study in detail(Slug Is A Surrogate Marker of Epithelial to Mesenchymal Transition (EMT) in Head and Neck Cancer, Steinbichler TB et al, J Clin Med; 2020 Jun 30;9(7):2061.) The percentage of Slug-positive cells correlated positively with the percentage of cytokeratin/vimentin double-positive cells (r = 0.41; R2= 0.17; p= 0.005). In Slug-positive tumors, 4.0 ± 2.6% of tumor cells were cytokeratin/vimentin double positive compared to 1.9 ± 1.8% in Slug negative tumors (p=0.001). Furthermore, Slug positive HNC specimens had a lower expression of epithelial E-cadherin (p<0.05) and β-catenin (p<0.05), a lower E-cadherin/β-catenin co-expression (p<0.05) and a higher vimentin/cytokeratin ratio(p=0.01) indicating simultaneous downregulation of epithelial and upregulation of mesenchymal markers consistent with EMT. These findings are now also included in the manuscript in more detail (page 12, line 10-23).
Authors need to perform IHC IgG control and normalize non-specific binding/background, which allows for better interpretation of specific staining at the antigen site.
Authors are thankful for that comment as it is absolutely correct. In all staining procedures isotype matching control immunoglobulins were used in the in the same final concentration as in the antibody staining conditions. The isotype controls never presented any visible reactions. This information was included into the methods section of the manuscript (page 5, line 30-32).
Did the authors determine Slug expression in cancer proliferating cells with or without treatment? Authors can perform co-expression with Ki67.
Both mRNA expression and western blot analysis were performed for Slug in several proliferating HNSCC cells lines, like SCC-25, UPCI-SCC090 and in FaDu cells. In all of these cell lines baseline Slug expression was detected, which increased after 1 ng/ml TGF-β1 treatment for 3+2 days. These data are part of a separate manuscript submitted elsewhere. Furthermore, co-expression analysis of Slug and Ki-67 was performed in 354 patient samples and was already published in the manuscript mentioned above. (Slug Is A Surrogate Marker of Epithelial to Mesenchymal Transition (EMT) in Head and Neck Cancer, Steinbichler TB et al, J Clin Med; 2020 Jun 30;9(7):2061.). Slug correlated positively with the scores for Ki-67 proliferation index (rho=0.15; p=0.005; n=354). These results were also included into the manuscript according to your suggestions (page 13, line 18-19).
Figure 1. Authors need to qualify the protein expression in IHC from Slug positive and negative samples and IgG control normalizes.
Authors are again grateful for that comment. The quantitative determination of Slug IHC has been detailed in a previous publication “Slug Is A Surrogate Marker of Epithelial to Mesenchymal Transition (EMT) in Head and Neck Cancer, Steinbichler TB et al, J Clin Med; 2020 Jun 30;9(7):2061.)” but was of course also included in the manuscript (page 3, line 3-9).
The authors need to provide a few representative IHC figures of Slug expression from treated group (table 2).
Figures of Slug IHC after first-line radiochemotherapy were included in the manuscript (Figure 3; page 9, line 12-16).
Did authors check Slug expression in HNSCC provided on the TCGA portal? Which could be the best tool to compare present observation with other studies?
We are currently working on external validation of Slug as predictive biomarker. Therefore, we evaluate the existing HNSCC genome data of the TCGA portal after having excluded patients with incomplete clinical data. Slug expression is assessed by Slug mRNA expression. Then -using a multivariable analysis- we are going to calculate whether „Slug positive“ patients have a significantly reduced long-term survival compared to „Slug negative“ patients when treated with first-line radiotherapy or radiochemotherapy.

Reviewer 2 Report
Cancers-1095802
Title: The epithelial-mesenchymal transcription factor Slug predicts survival benefit of up-front surgery in head and neck cancer
Journal: Cancers
In this manuscript the authors have investigate if overexpression of Slug can be used as a predictive biomarker for radio- and chemotherapy resistance.
In conclusion, they show that Slug-IHC could be a predictive biomarker to support upfront surgery.
It is an interesting study, well designed, well written but still with some weaknesses.
Comments and questions:
- In this study the authors study Slug, p16, p53, PD-L1, CA9 and ERCC1 expression in tumor biopsies but only the Slug results are well described. The results about p16, p53, ERCC1 and CA9 must been shown more in detail in Supp.Tables.
- The results are shortly described in the Result part about the correlation with above mentioned proteins and Slug expression. However here they say that p53 positive correlate significantly with Slug but in the discussion part they say that p53 overexpression and total absence of p53 correlate. Why different?
- Why is T1-3 in the same group? It is a very big difference between T1 and T3 tumors.
- Slug is one of many transcription factor för EMT in HNSCC and it is well known that different tumors have overexpression of different transcription factors for EMT.
So, one important question here is if EMT is important or is Slug the important factor for better survival? To strengthen the result at least two more transcription factors for EMT has to be investigated in the same patient cohort.
Author Response
Object: Manuscript ID: cancers-1095802; Type of manuscript: Article; Title: The epithelial-mesenchymal transcription factor Slug predicts survival benefit of up-front surgery in head and neck cancer; Authors: Herbert Riechelmann, Teresa Bernadette Steinbichler *, Susanne Sprung, Annette Runge, Ute Ganswindt, Gabriele Gamerith, Jozsef Dudas; Received: 16 January 2021
Author’s response to review
Thank you for consideration of our manuscript for publication in your journal.
We have reviewed the manuscript according to the reviewer’s and the editor’s comments carefully.
Reviewer #2 (Comments to the Author (Required)):
In this manuscript the authors have investigate if overexpression of Slug can be used as a predictive biomarker for radio- and chemotherapy resistance.
In conclusion, they show that Slug-IHC could be a predictive biomarker to support upfront surgery.
It is an interesting study, well designed, well written but still with some weaknesses.
Thank you for your thorough review of our manuscript. We tried to answer all your comments carefully and revised the manuscript accordingly.
Comments and questions:
- In this study the authors study Slug, p16, p53, PD-L1, CA9 and ERCC1 expression in tumor biopsies but only the Slug results are well described. The results about p16, p53, ERCC1 and CA9 must been shown more in detail in Supp.Tables.
Unfortunately, we have already published the co-expression of Slug and the other mentioned biomarkers in another study in more detail (Slug Is A Surrogate Marker of Epithelial to Mesenchymal Transition (EMT) in Head and Neck Cancer, Steinbichler TB et al, J Clin Med; 2020 Jun 30;9(7):2061.) Nevertheless, we tried to meet your comment and included more information on these other biomarkers in the manuscript (page 7, line14-17). Slug-positive patients were more frequently p16 negative (rho=-0.13; p<0.001; n=354), but ERCC1 (rho=0.16; n=45; p<0.005), CAIX (rho=0.29; p<0.001; n=175), CD44 (rho=0.18; p = 0.02; n = 160), MMP9 (rho=0.19; p<0.001; n=352) and p53 (rho=o.27; n=102; p<0.001) positive. Slug expression was not related to PD-L1 expression (p=0.89).
- The results are shortly described in the Result part about the correlation with above mentioned proteins and Slug expression. However here they say that p53 positive correlate significantly with Slug but in the discussion part they say that p53 overexpression and total absence of p53 correlate. Why different?
According to a previous publication by Bouchalova and colleagues scattered p53 staining is considered to be associated with regular genetic background without nonsilent mutations, while no staining at all is related with complete loss of p53 protein, due to deletions. Increased (over 66% of tumor cells stained) staining pattern is considered to be associated with nonsilent mutated p53 (Bouchalova P.; Nenutil, R.; Muller, P.; Hrstka, R.; Appleyard, M.V.; Murray, K.; Jordan, L.B.; Purdie, C.A.; Quinlan, P.; 610 Thompson, A.M., et al. Mutant p53 accumulation in human breast cancer is not an intrinsic property or dependent on 611 structural or functional disruption but is regulated by exogenous stress and receptor status. J Pathol 2014, 233, 238-246, 612 doi:10.1002/path.4356). Consequently, we described p53 positivity (positive for nonsilent mutated p53) as either overexpression or total absence of p53. We amplified the complete protein coding region of p53 mRNA and sequenced it. Thereby we found a statistically significant correlation between confirmed p53 sequence mutations or mRNA loss and irregular staining pattern. Irregular gene expression consists of sequencing confirmed mutations and lack of gene product, which was also confirmed by PCR and sequencing. A scattered, regular p53 staining pattern and wild type p53 mRNA sequence were also related (Spearman R: 0.617; p<10-4). These data are part of separate manuscript submitted elsewhere. However, we are grateful for your comment as this might be very confusing for the reader so we clarified this in more detail in the methods section (page 5, line 20-26).
- Why is T1-3 in the same group? It is a very big difference between T1 and T3 tumors.
Authors are grateful for that comment. There are several reasons for this:
1) To compare groups, you have to stratify. However, overstratification should be avoided.
2) T1-T3 vs. T4 has roughly the same number of cases in this patient group
3) T1-T3 is limited to the organ of origin of the tumor while T4 extends beyond organ borders
4) In accordance with cancer registry practices, N is encoded separately, so N0 and T1-T3 is localized disease and any N including N0 and T4 is regional disease.
This is now also mentioned in the Discussion section of the manuscript (page 12, line 12-16).
- Slug is one of many transcription factors for EMT in HNSCC and it is well known that different tumors have overexpression of different transcription factors for EMT. So, one important question here is if EMT is important or is Slug the important factor for better survival? To strengthen the result at least two more transcription factors for EMT has to be investigated in the same patient cohort.
Immunohistochemistry with specific antibody for Snail (reactive with an epitope not present in Slug) did not reveal any reaction in Slug-positive HNSCC. Furthermore, we evaluated mRNA expression for Twist and ZEB1 in 10 control and 37 head and neck cancer tissue biopsies. These EMT-related transcriptional factors had a low overall expression and showed no significant difference of gene expression in HNSCC related to normal mucosa. Consequently, we think that Slug is the major EMT-transcriptional factor in HNSCC and we decided to use Slug as a clinical indicator for EMT in HNSCC in this study. These results are now also mentioned in the manuscript (page 13, line 28-32).

Round 2
Reviewer 1 Report
Thanks for addressing the questions. The authors have suitably addressed and explained all the questions raised in the previous version of the manuscript. I am hereby recommending the current form of this manuscript to consider for publications.
Reviewer 2 Report
I am satisfied with the authors answers and changes in the manuscript.